# Knowledge, attitudes, and practices against the growing threat of COVID-19 among medical students of Pakistan

Khola Noreen[1]*, Zil-e- Rubab[2], Muhammad Umar[3], Rehana Rehman[4], Mukhtiar Baig[5], Fizzah Baig[6]

1 Department of Community Medicine, Rawalpindi Medical University, Rawalpindi, Pakistan, 2 Ziauddin Medical College-Ziauddin University, Karachi, Pakistan, 3 Rawalpindi Medical University, Rawalpindi, Pakistan, 4 Department of Biological and Biomedical Sciences, Aga Khan University, Karachi, Pakistan, 5 Department of Clinical Biochemistry, Faculty of Medicine, Rabigh, King Abdulaziz University, Jeddah, Saudi Arabia, 6 Ziauddin Medical College-Ziauddin University, Karachi, Pakistan

* khauladr@gmail.com

## Abstract

### Background

On account of the COVID-19 pandemic, many changes have been implicated in university medical students. We are cognizant that pandemic can be controlled with dedicated contributions from all involved in the healthcare profession. Therefore, it is important to know the pandemic and application of knowledge by the medical students to formulate a further line of management in Pakistan.

### Objective

We aimed toinvestigate the knowledge, attitudes, and practices (KAP) regarding COVID-19 and its impact on medical students of Pakistan.

### Methods

A cross-sectional survey was conducted in June 2020 by a validated self—administered questionnaire. The survey instrument was tailored from a published questionnaire comprised of questions on demographics (6), knowledge (14), attitudes (4), and practices (6).

### Results

Of the 1474 medical students in the study, 576(39.1%) were males, and 898(60.1%) were females. Two-thirds of the participants 1057(71.7%) had adequate knowledge, and almost all the students had positive attitudes (1363(92.5%), and good practices 1406(95.4%) to COVID-19. Two-thirds of the medical students 1023(69%) believed that the COVID-19 outbreak had affected their social, mental, and psychological well-being. One-quarter of the medical students 391(26%) become more religious, 597(40%) realized the importance of life, and 1140(77%) became careless because of the pandemic. The female medical students were 2.545 times (p < .001) and 4.414 times (p < .001) more likely to have positive attitudes and good practices toward COVID-19 as compared to males.

**Data Availability Statement:** All relevant data are within the paper and its Supporting Information files.

**Funding:** "The author(s) received no specific funding for this work."

**Competing interests:** "The authors have declared that no competing interests exist."

## Conclusion

Medical students, especially females and senior year scholars, were well-versed with desired levels of knowledge, attitudes, and preventive measures toward COVID-19. Most of them recognized COVID 19, is shaping their social, mental, and psychological well-being and encroaching on the healthcare system and economy. The information acquired by the KAP study may help to devise effective preventive strategies for future events.

## Introduction

The world has undergone drastic changes in the deadly virus's unprecedented emergence, namely Novel Coronavirus (nCOVID-19). Since the report of the first cluster of Coronavirus cases on 31 December 2019 in Wuhan, a metropolitan city of China, it has shown rapid spread over a short period [1]. On 30 January 2020, the International Health Regulations Emergency Committee meeting regarding the outbreak of COVID-19 declared it a Global Public Health Emergency of International Concern (PHEIC). On 11 February 2020, the virus was labelled by WHO as 'severe acute respiratory tract coronavirus-2' (SARS-CoV-2)and diseases COVID-19 [2].On 11 March 2020, this outbreak was declared as *Global Pandemic* by WHO [3]. By that time, more than 118,000 cases were reported in 114 countries, and 4,291 people have lost their lives [4].

As of 4 July 2020, 11,191,676 cases and 529,127 deaths have been reported globally [5]. Europe is the most affected region with (2,638,903) cases and the highest number of deaths (196,169). The African region is least affected with 268,102 cases and 5,673 deaths [2].

In Pakistan first two cases of COVID-19 were confirmed by the Ministry of Health, government of Pakistan on 26 February 2020. The first case was confirmed in Karachi, and other in Islamabad, both have a history of travelling to the Islamic Republic of Iran [6]. Till today (4July 2020), 221,896 active cases, and 4,551 deaths have been reported in Pakistan [7].

The SARS-CoV outbreak occurred in China in 2002, spread to 17 countries, and infected 8,089 people with a case fatality rate of 9.6% [8]. While MERS-CoV occurred in Saudi Arabia in 2012, it spread to 21 countries and infected 2,506 people with a 34% case-fatality rate [9]. Case fatality rate of SARS-CoV-2 is 4.7%(11,191,676 cases &529,127 deaths on 4 July 2020) [5].

Novelty of virus, lack of evidence-based management guidelines, scarcity of scientific literature available regarding its epidemiology, virulence, infectivity, mode of transmission, prevention, and management has resulted in the spread of fake and unauthentic news leading to chaos and uncertainty [10]. Social and electronic media become an indispensable real-time source of information. *The WHO Director-General stated*: "*We're not just fighting an epidemic; we're fighting an infodemic.*" Every day social media networking sites, blogs, and different forums are abuzz with newsflash about an increase in disease mortality, human suffering, reports of economic collapse, and atrocious stories about deserted streets in busiest and crowded areas of the world [11].

Social media is the major source of spreading different myths regarding prevention and resistance against the disease. The knowledge gap can create deleterious effects during the pandemic as it can add to stress and chaos. Moreover, negative attitudes and practices, various misconceptions, and myths can exacerbate its catastrophic consequences. Evidence from previous epidemics, MERS and SARS, has shown that assessment of the knowledge, attitudes, and practices help in identifying myths, taboos, and misinformation related to epidemic and help to devise effective strategies to mitigate its deleterious effects [12].

A global effort to identify effective management options to deal with the COVID-19 pandemic is in full flow. However, potential medications remain under investigation, and vaccines are unlikely to be available shortly. According to KAP theory, society's promptness to accept behavioral change and adherence to preventive strategies is largely influenced by knowledge, attitudes, and practices [13]. Successful implementation of preventive strategies can be achieved by increasing knowledge about transmission mode, creating awareness about preventive strategies, catering for myths and misconceptions and developing positive attitudes to adopt healthy hygienic practices that can help to mitigate its deleterious effects of deadly virus [12].

There is not much information about Coronavirus among medical students regarding knowledge, attitudes, and preventive practices. Therefore, we aimed to investigate the knowledge, attitudes, and practices regarding COVID-19 and its impact on Pakistani medical students.

## Methods

This survey was conducted among medical students from different Pakistani universities, from 13 June to 29 June 2020. Since the current pandemic situation has resulted in the suspension of routine academic activities, restricted movement, social distancing, and isolation, so considering current circumstances the data was collected using an online questionnaire developed by means of Google form. The questionnaire was disseminated to undergraduate medical students of Pakistan's leading medical universities via SM platforms like Facebook, Twitter, WhatsApp's, and Instagram. Students were also approached by emails, and personal contacts. A non-probability convenient sampling technique was employed for the recruitment of study participants. The sample size was calculated on Raosoft sample size calculator. It was 658 to achieve the confidence level of 99% with a margin of error of 5%, response distribution 50%, and the population size of 70,000 for the present survey. It was intended to approach as many students as possible to gather maximum possible data to enhance the study's validity and generalizability.

### Data collection tool

The self-reported questionnaire was originally developed by an extensive literature review of already published literature and WHO myth-buster document [14]. Already available literature was explored in-depth and synchronized into a conceptual framework and grouped various questions under the different themes, including knowledge, attitudes, practices, and misconceptions [15, 16].

### Validity and reliability of the study tool

Two senior faculty members were requested to review the tool for its construct and content validity. Items that need exclusion were highlighted, omissions of repetition were done, the double-barrel questions were removed, discrepancies were rectified, a rephrasing of long statements was done to make them simple, clear, and unambiguous. A pilot study was conducted and the tool was administered to 40 students to check its understanding and reliability. Its Cronbach's alpha was 0.79.

### Data collection/recruitment procedure

This online survey form was electronically shared, and data was collected voluntarily, and the consent statement was included at the beginning of the online questionnaire. All participants

were bound to give their willingness to volunteer participation, and filling the questionnaire was considered their consent before proceeding further. A brief description of the study questionnaire, purpose of study, and instructions to fill the questionnaire was given before filling the survey. The willingness of participants was sought by giving them the option of yes and No. If they showed a willingness to proceed by selecting "yes "then they were allowed to access the detailed questionnaire on the next page to complete and submit it online, those who opted *"No"* option were not allowed to proceed. Moreover, participants were given the option to withdraw anytime if they were not willing to proceed further. The invitation for participation was sent to different medical colleges/universities. A total of 1800 medical students were approached, complete responses were acquired from 1474 medical students with a response rate of 82%.

## Study questionnaire scoring

"The questionnaire comprised of five parts; (1) demographics, which surveyed participants' socio-demographic information, including gender, age, academic year, marital status occupation, and parental income; (2) knowledge about COVID-19 (K1-14); (3) attitudes toward COVID-19(A1-A4); and (4) practices relevant to COVID-19(P1-P6); (5) impact questions."
"Questions related to KAP had three options "true/false/not sure. There were fourteen knowledge questions, and for each item, one score was given for true and zero for false and not sure. An individual score of 1–10 was taken as inadequate, while the score in the range of 11–14 was counted adequate." "Four questions related to attitudes, and the scores were awarded +1 for true and -1 for false and not sure. So the total score ranged from -4 to +4. The plus scores were taken as positive attitudes, while negative scoring indicated negative attitudes." "For practice questions, 2 points were awarded for yes, one for sometimes and zero for no, and the scores = >6were taken as adequate and < 6, were taken as inadequate."

## Data processing and statistical analysis

The SPSS-26 was employed to analyze the data. Frequency and percentages were computed for the categorical variables. A chi-square test was performed for investigating the comparison between different categorical variables. Binary logistic regression analysis was used to explore the association of knowledge, attitudes, and practices score with gender and academic years. All p-values <0.05 were considered significant.

## Ethical approval

The data collection procedure complies with institutional and National ethical guidelines and following the Helsinki declaration. Anonymity and confidentiality of data were maintained. The study was carried out after obtaining ethical approval from the Institutional Review Board of Rawalpindi Medical University(Reference No. 88/1REF/RMU/202).

## Results

A total of 1474 medical students (576[39.1%] males, and 898[60.1%] females] were included in the study. The general characteristics of the participants' are shown in Table 1.

One-third of the medical students did not know "SARS-CoV-2 causes COVID-19 infection," and "all community members are equally at risk for COVID-19." Surprisingly, half of the students didn't believe that the "risk of getting infected when traveling by plane is higher." About 39% of the students thought that the "virus is human-made and deliberately released."

**Table 1. General characteristics of study participants.**

| Variables | | N | % |
|---|---|---|---|
| Gender | Male | 576 | 39.1 |
| | Female | 898 | 60.9 |
| Academic year | First year | 252 | 17.1 |
| | Second year | 252 | 17.1 |
| | Third year | 185 | 12.6 |
| | Fourth year | 706 | 47.9 |
| | Fifth year | 79 | 5.4 |
| Marital Status | Married | 30 | 2 |
| | Unmarried | 1438 | 97.6 |
| | Divorced | 6 | .4 |

The medical students' showed positive attitudes and good practices against COVID-19 (Table 2).

The majority of participants 1201(81.5%) seek information regarding COVID-19 from television, 687(46.65) from SM, 672(45.6%) from newspapers and other sources (Fig 1).

**Table 2. Study participants knowledge, attitudes, and practices regarding COVID-19 pandemic.**

| Questions | Statements | Frequency | Percentage |
|---|---|---|---|
| | Knowledge | Correct Answer | |
| K1 | COVID-19 infection is caused by SARS-CoV-2. | 987 | 67 |
| K2 | COVID-19 infection is spread via respiratory droplets of the infected person. | 1422 | 96.5 |
| K3 | All community members are equally at risk for COVID-19. | 999 | 67.8 |
| K4 | The best way of preventing spread of COVID-19 is social distancing | 1419 | 96.3 |
| K5 | The virus is human-made and deliberately released | 579 | 39.3 |
| K6 | Any type of group activity may spread this infection | 1324 | 89.8 |
| K7 | A symptomless COVID-19 patient (during incubation period) can't transmit infection | 1146 | 77.7 |
| K8 | The risk of getting infected when travelling by plane is higher | 725 | 49.2 |
| K9 | This virus infection can be avoided by frequent hand washings by soap | 1410 | 95.7 |
| K10 | Advising Quarantine to passengers coming from infected areas is a good practice to avoid spread of infection | 1446 | 98.1 |
| K11 | Lockdown all over the country will control the spread of this virus | 1292 | 87.7 |
| K12 | Closing teaching institutions and shopping malls are effective ways of social distancing | 1377 | 93.4 |
| K13 | The most common cause of spread of this infection in any country is traveler from infected area | 1174 | 79.6 |
| K14 | Isolation period for infected people and those exposed to infection is 14 days | 1204 | 81.7 |
| | Attitude Questions | True Answer | |
| A1 | I am sure that COVID-19 infection will be overcome soon. | 643 | 43.6 |
| A2 | We can overcome this problem by taking precautionary steps | 1379 | 93.6 |
| A3 | I understand that this infection is highly contagious | 1407 | 95.5 |
| A4 | It is my social responsibility to take safety measures in controlling spread of this infection. | 1458 | 98.9 |
| | Practice questions | Yes | |
| P1 | I am avoiding meeting my friends and relatives | 1167 | 79.2 |
| P2 | I am avoiding visiting crowded place | 1352 | 91.7 |
| P3 | I am avoiding using ATM machine. | 1131 | 76.7 |
| P4 | I prefer to walk by stairs then using lift | 1199 | 81.3 |
| P5 | I am using face mask outside the home | 1269 | 86.1 |
| P6 | I am using soap frequently for handwashing | 1351 | 91.7 |

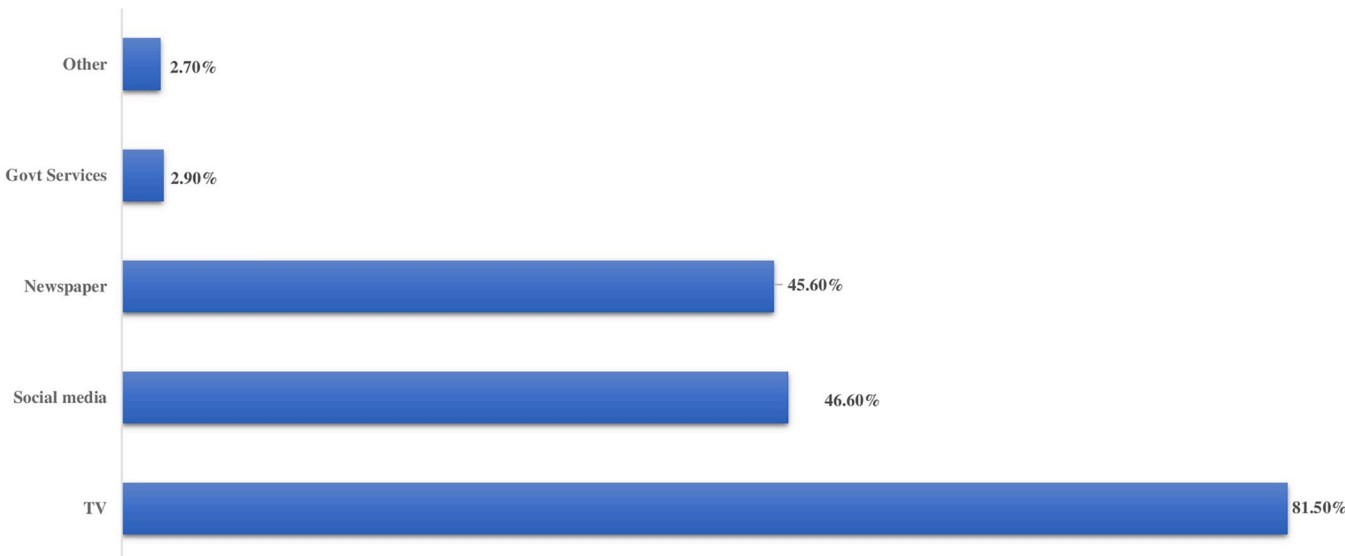

**Fig 1. Sources of information regarding COVID-19.**

Two-thirds of the participants 1057(71.7%) had adequate knowledge, and almost all the students had positive attitudes (1363(92.5%), and good practices 1406(95.4%) to COVID-19 (Fig 2).

Responses of the medical students regarding the impact of the COVID-19 pandemic were variable. Two-thirds of the medical students 1023(69%) believed that the COVID-19

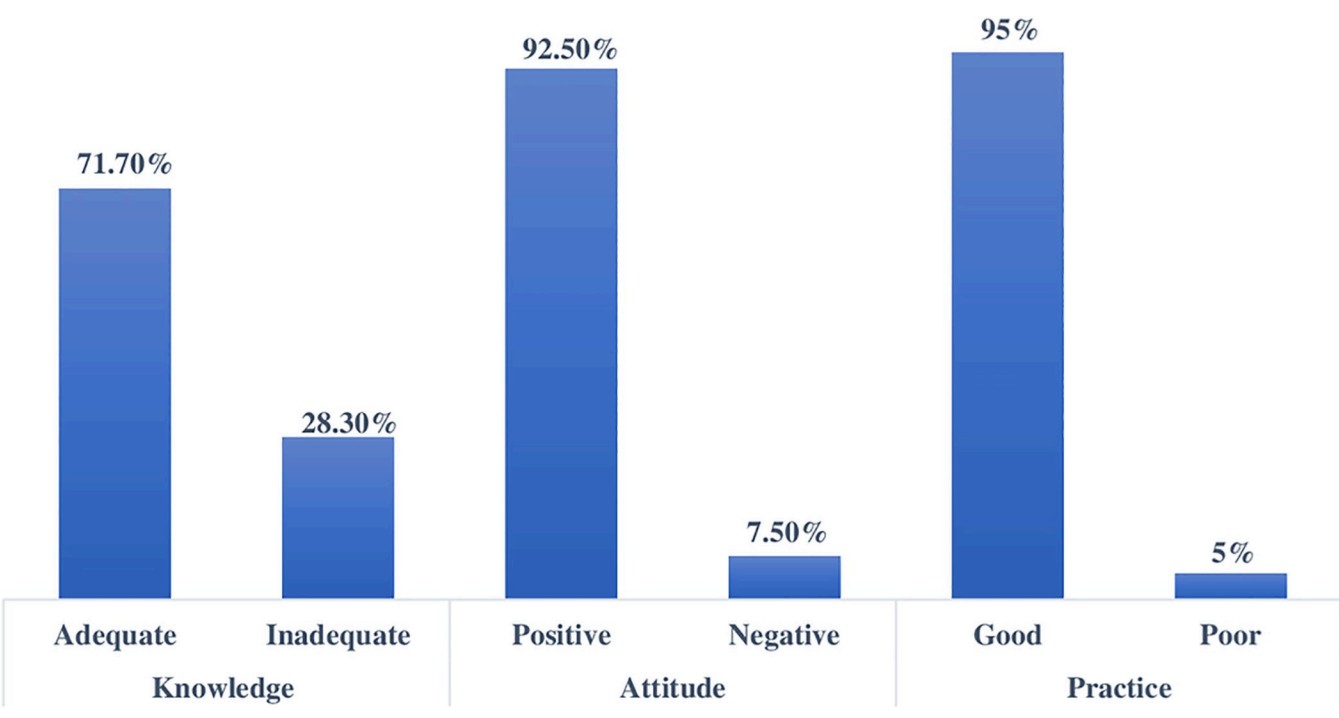

**Fig 2. Medical students' knowledge, attitudes, and practices categories.**

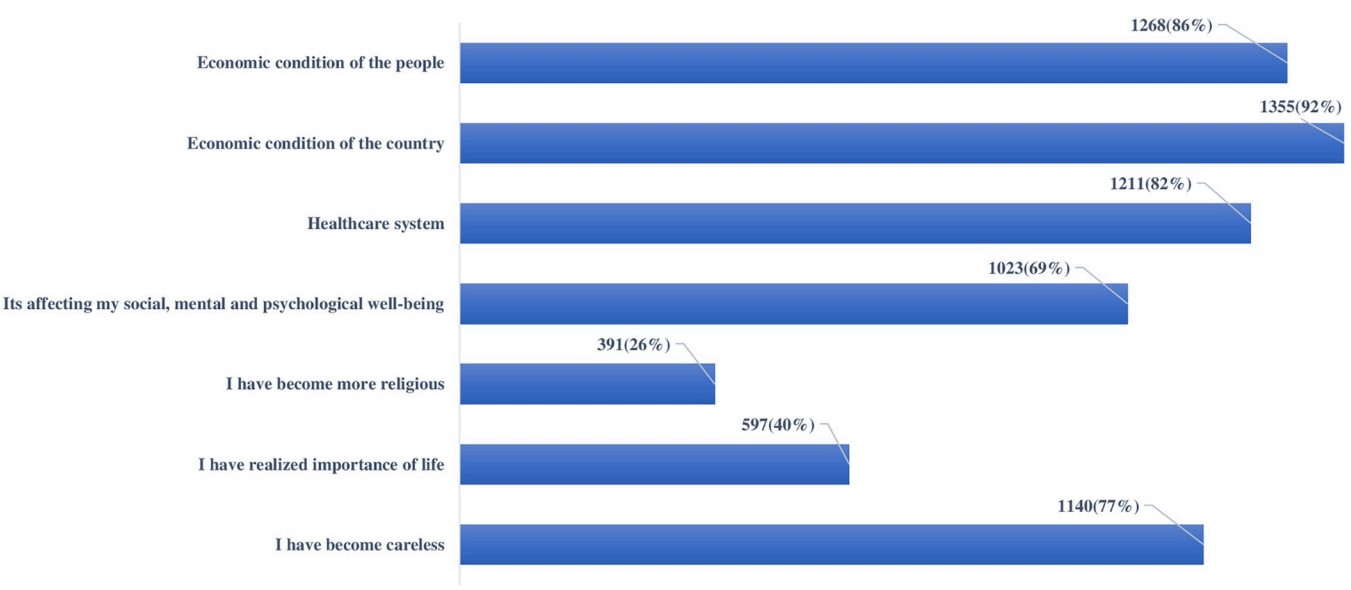

**Fig 3. Impact of COVID-19 pandemic.**

outbreak affected their social, mental, and psychological well-being. One-quarter of the medical students 391(26%) became more religious, 597(40%) realized the importance of life, and 1140(77%) became careless because of the pandemic. The majority of the students believed that COVID-19 is affecting the healthcare system and economic condition of the people and country (Fig 3).

Females medical students had adequate knowledge, positive attitudes, and good practices against COVID-19 than males. The fourth and fifth-year students had adequate knowledge scores compared to other groups, while the fourth-year students had good practices as well (Table 3).

The logistic regression analysis showed that female medical students were 1.367 times (p < .001) more likely to have adequate knowledge of COVID-19 than male students. Similarly, the female medical students were 2.545 times (p < .001) and 4.414 times (p < .001) more likely to have positive attitudes and good practices toward COVID-19 compared to males (Table 4).

**Table 3. Comparison of knowledge, attitude and practice scores according to different variables.**

| Variables | | Knowledge | | Attitudes | | Practices | |
|---|---|---|---|---|---|---|---|
| | | Inadequate N (%) | Adequate N (%) | Negative N (%) | Positive N (%) | Poor N (%) | Good N (%) |
| Gender | Male | 188(32.6) | 388 (67.4) | 67(11.6) | 509 (88.4) | 49(8.5) | 527(91.5) |
| | Female | 229(25.5) | 669(74.5) | 44 (4.9) | 854(95.1) | 19(2.1) | 879(97.9) |
| | p-value | 0.003* | | < 0.001* | | < 0.001* | |
| Education level | First year | 98 (38.9) | 154 (61.1) | 25(9.9) | 227 (90.1) | 15(6) | 237(94) |
| | Second year | 97(38.5) | 155 (61.5) | 18(7.1) | 234(92.9) | 15(6) | 237(94) |
| | Third year | 54(29.2) | 131(70.8) | 12 (6.5) | 173 (93.5) | 16(8.6) | 169 (91.4) |
| | Fourth year | 154(21.8) | 552(78.2) | 56(7.9) | 650 (92.1) | 21(3) | 685(97) |
| | Fifth year | 14(17.7) | 65(82.3) | 0(0) | 79(100) | 1(1.3) | 78(98.7) |
| | p-value | <0.001 | | 0.061 | | 0.004 | |

**Table 4. Association of knowledge, attitude and practice scores with different variables (binary logistic regression analysis).**

| Variables | Knowledge | | | | | Attitudes | | | | | Practices | | | | |
|---|---|---|---|---|---|---|---|---|---|---|---|---|---|---|---|
| | B | P-value | OR | 95% CI | | B | P-value | OR | 95% CI | | B | P-value | OR | 95% CI | |
| | | | | Lower Bound | Upper Bound | | | | Lower Bound | Upper Bound | | | | Lower Bound | Upper Bound |
| Gender (Female) | .312 | .009 | 1.367 | 1.081 | 1.728 | .934 | .000 | 2.545 | 1.708 | 3.794 | 1.485 | .000 | 4.414 | 2.553 | 7.632 |
| First year | -1.064 | .001 | .345 | .184 | .649 | -18.91 | .997 | .000 | .000 | . | -1.499 | .152 | .223 | .029 | 1.734 |
| Second year | -1.010 | .002 | .364 | .193 | .686 | -18.44 | .997 | .000 | .000 | . | -1.335 | .202 | .263 | .034 | 2.046 |
| Third year | -.643 | .056 | .526 | .272 | 1.017 | -18.49 | .997 | .000 | .000 | . | -2.000 | .056 | .135 | .017 | 1.050 |
| Fourth year | -.234 | .448 | .791 | .432 | 1.450 | -18.66 | .997 | .000 | .000 | . | -.756 | .465 | .470 | .062 | 3.567 |

## Discussion

A significant number of deaths within a short period ascertained SARS-CoV-2, a dire menace to global public health. Public health education is distinguished as an efficient measure to combat public health emergencies by organizing the public against such untoward conditions. It may influence society's KAP by extending proper knowledge, moderating panic, and reassuring optimistic attitudes and keeping the public complying with new practices. All these KAP elements are crucial to ensure effective prevention and control of the emergency.

The present cross-sectional survey results showed Pakistani medical students were well aware of current knowledge along with optimistic attitudes toward COVID-19. Medical Students were also very conscious of counter regulatory measures. The majority of the students, especially senior years were adequately cognisant with COVID-19 related knowledge indicating that effective health education has been delivered either by television through the massive public education campaigns, SM, and other sources. These results are consistent with H1N1 related KAP among university students in South Korea, UK, and Hong Kong [17–19] Medical undergraduates' involvement in delivering patients' care, merged with the exponential growth of diseases resulting in pandemics, places the population's current cluster at greater risk for catching and transferring the disease. During previous and current pandemics such as influenza and COVID-19, healthcare delivery departments are laid beneath the huge burden. A scarcity of healthcare professionals may determine the contribution of under training health professionals such as medical undergraduates. Moreover, medical undergraduates are generally consigned for healthcare assistance from households and families. They have expounded improved knowledge than non-medical undergraduates [20], which supposedly is better in senior years medical students [21].

Our evaluation of the informants used by medical undergraduates to ascertain about COVID-19 disclosed predicted substantial dependence on SM and television. This is in accordance with a related study conducted in Turkish university where information from social media was the source for learning about the influenza pandemic [22], but contrary to a study on less addressed subject such as the Zika virus epidemic where news channels appeared to be the chief source of information [23].

Our findings can guide policymakers to the significance of SM in publicizing evidence to the community, particularly in pandemics. The approved sites, such as the (covid.gov.pk) and medical search engines like PubMed, were not frequently used than social websites and news channels to acquire information. Our data suggest a need to improve the discernibility of reliable sources of information, even within a small cluster of populations that should be more conversant than the public with unswerving medical websites.

The current survey also showed that the female medical students had adequate knowledge, positive attitudes, and good practices against COVID-19 than males, which is in agreement with the result of a Saudi Arabian study on MERS [24]. A few other studies also stated that females were more sensitized in practicing hand hygiene and wearing protective masks in context to infectious diseases like H1N1, SARS, and MERS [18, 25–27]. Medical students' training can also justify the significantly high scores in the knowledge they gained in clinical setups. Their consciousness of duty and concern as a future medical professional may also motivated them to show more positive attitudes and proactive practices during this pandemic [28].

A Pakistani cross-sectional study has documented good knowledge, positive attitude, and reasonable practices regarding COVID-19 infection inprimary healthcare providers from three tertiary care hospitals [29]. In contrast to our results, a Pakistani study reported low knowledge scores among the general public, including students [30]. Research on medical students from Iran indicated a high level of related knowledge and high performance in preventive behaviors but moderate risk perception [31]. In Jordan, medical and non-medical students obtained an average knowledge score. The majority of the participants had good knowledge of COVID-19 symptoms and were aware of the lack of vaccine and treatment for COVID-19 [32].

Together, these results recommend that gender and education areas like medicine potentially affect students' responses to the COVID-19 pandemic and public health education attainment. This should be taken into account by educational and healthcare authorities. These factors should also be considered when they devise exigency plan or train them against similar public health calamities.

The impact of COVID-19 is enormous, especially for under-developed countries. The long term lockdown is not the solution, and shopping malls, air routes, and borders cannot be kept closed for longer periods [33]. Therefore, currently and in the future, the KAP toward COVID-19 will show a fundamental part in defining people's promptness to adopt behavior alteration steps from health experts. KAP studies postulate standard evidence to establish the type of intercession that can be compulsory for amendment of misapprehensions about the virus. Evaluating the current KAP related to COVID-19 among the medical students will be useful in delivering improved insight to address knowledge about the disease and the improvement of preventive strategies. It is suggested that health campaigns involve senior medical students; thus, the health care system's burden would be distributed, and the disease would be contained faster. With limited resources, countries like Pakistan should apply policies to keep their medical undergraduates rationalized about evolving public health and medical predicaments. Medical students should also be appropriately directed to valid informants during these periods. Given the current global situation, frequent SM deployment by medical colleges to increase knowledge is inevitable. Strategies should be designed to apply such propagation in the initial stages of medical and public health disasters.

## Implications of the findings

As a response to the COVID-19 pandemic majority of study participants embraced social isolation strategies, regular hand washing, and improved personal hygiene procedures as their initial line of defense against the COVID 19. Pakistani medical students presented a predictable level of knowledge about the COVID-19 and implemented appropriate strategies to stop its spread.

Moreover, dedicated training programs for medical professionals at the Government level can supplement their knowledge of risks and preventive strategies related to COVID-19, which will help them deliver proper care to their patients and keep themselves safe from the

virus. Recently, the WHO Director-General's opening remarks at the media briefing on COVID-19acclaimed Pakistan in deploying the infrastructure built up over many years for polio to combat COVID-19. Community health workers who have been trained to go door-to-door vaccinating children for polio have been utilized for surveillance, contact tracing, and care. Optimistically, by scheming effective COVID-19 prevention and management programs at the Government level, countries like Pakistan can manage the spread of COVID-19 infection. A worldwide public health driven strategy to improve knowledge should be conducted to combat the disease. In this regard, everyone should put efforts to eradicate this pandemic.

## Limitations

This study's results provide baseline data about KAP towards COVID -19 pandemic, which will help devise effective preventive strategies for future events. However, this study has certain limitations; the data was collected using a self-reported questionnaire, which can be a potential cause of reporting bias. Moreover, since data is collected from medical students, so there is a possibility that they might answer the question positively on basis of their medical knowledge as they already perceive what is expected from them. Another limitation is that study participants' were enrolled using a non-probability convenience sampling technique, and data was collected online through social networking platforms. There is a possibility of bias as we may not be able to approach the students with an internet connectivity issue.

## Conclusion

Generally, medical students, especially females and senior year scholars, were well-versed with desired levels of knowledge, attitude, and preventive measures towards COVID-19.

Most of them recognized COVID 19, shaping social, mental, and psychological well-being and encroaching the healthcare system and economy.

## Supporting information

**S1 Data.**
(SAV)

## Author Contributions

**Conceptualization:** Muhammad Umar, Rehana Rehman.

**Data curation:** Khola Noreen, Mukhtiar Baig, Fizzah Baig.

**Formal analysis:** Khola Noreen, Mukhtiar Baig.

**Investigation:** Khola Noreen, Rehana Rehman.

**Methodology:** Khola Noreen.

**Project administration:** Rehana Rehman.

**Resources:** Khola Noreen.

**Software:** Mukhtiar Baig.

**Supervision:** Muhammad Umar.

**Writing – review & editing:** Zil-e- Rubab.

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
