## [Decision Letter · Decision Letter 0]

10 Sep 2020

PONE-D-20-23756

Knowledge, Attitudes, and practices against the growing threat of COVID-19 among medical students of Pakistan

PLOS ONE

Dear Dr. Noreen,

Thank you for submitting your manuscript to PLOS ONE. After careful consideration, we feel that it has merit but does not fully meet PLOS ONE’s publication criteria as it currently stands. Therefore, we invite you to submit a revised version of the manuscript that addresses the points raised during the review process.

We look forward to receiving your revised manuscript.

Kind regards,

Ramesh Kumar, PhD

Academic Editor

PLOS ONE

Journal Requirements:

2.We note that you have indicated that data from this study are available upon request. PLOS only allows data to be available upon request if there are legal or ethical restrictions on sharing data publicly. For information on unacceptable data access restrictions, please see http://journals.plos.org/plosone/s/data-availability#loc-unacceptable-data-access-restrictions.

3. Please ensure that you include a title page within your main document. You should list all authors and all affiliations as per our author instructions and clearly indicate the corresponding author.

4. Please remove your figures from within your manuscript file, leaving only the individual TIFF/EPS image files, uploaded separately.  These will be automatically included in the reviewers’ PDF.

Reviewers' comments:

Reviewer's Responses to Questions

**Comments to the Author**

1. Is the manuscript technically sound, and do the data support the conclusions?

Reviewer #1: Partly

Reviewer #2: Yes

2. Has the statistical analysis been performed appropriately and rigorously? 

Reviewer #1: No

Reviewer #2: Yes

3. Have the authors made all data underlying the findings in their manuscript fully available?

Reviewer #1: Yes

Reviewer #2: Yes

4. Is the manuscript presented in an intelligible fashion and written in standard English?

Reviewer #1: Yes

Reviewer #2: Yes

5. Review Comments to the Author

Reviewer #1: Good effort by the investigators. However, there are some major concerns with the methods and analysis that need to be addressed.

1.How were the main outcomes of knowledge and practice adequacy defined? Please provide rationales for choosing the cut-off points for positive attitudes and adequate practice.

2.What was the response rate? How many medical students were invited to participate vs those who participated in the survey? This has implications on the generalizability of findings to within Pakistan.

3.Please provide 95% CI for the OR in Table 4. As mentioned above, please clarify whether the models have adjusted for any variables.

4.How do the findings about factors associated with COVID-19 related knowledge and practice adequacy in Pakistan compare with literature from elsewhere?

5.In the discussion, please add implications of the findings and what steps are needed to address the gaps identified?

Other comments:

Please proofread and correct grammatical errors throughout the paper.

Please make the format of the paper consistent (font, color, line spaces) and provide the complete tables.

Reviewer #2: This is an intersting paper and is written at time when it is needed. Furthermore, this manuscript is written on issue which is very relevant in todays world. I have no comment to make hence ask for acceptance.

6. PLOS authors have the option to publish the peer review history of their article (what does this mean?). If published, this will include your full peer review and any attached files.

Reviewer #1: No

Reviewer #2: **Yes: **Faisal Abbas

---

## [Author Response · Author response to Decision Letter 0]

26 Sep 2020

Reply to comments

Reply: Rebuttal letter is being uploaded as a separate file.

Reply: The revised manuscript is being uploaded as a separate file as 

an unmarked version of your revised paper without tracked changes. 

You should upload this as a separate file labeled 'Manuscript'.

Reply: Unmarked manuscript is being uploaded as a separate file.

Reply: We have modified the style of our manuscript as required by the Plos One.

 Reply: 

Reply: SPSS data sheet is being uploaded.

3. Please ensure that you include a title page within your main document. You should list all authors and all affiliations as per our author instructions and clearly indicate the corresponding author.

 Reply: The title page has been included in the main document.

4. Please remove your figures from within your manuscript file, leaving only the individual TIFF/EPS image files, uploaded separately. These will be automatically included in the reviewers’ PDF.

Reply: Figures have been separated and have been converted to TIFF.

Reviewers' comments:

Reviewer's Responses to Questions

Comments to the Author

1. Is the manuscript technically sound, and do the data support the conclusions?

Reviewer #1: Partly

Reviewer #2: Yes

Reply: Thank you for your comments.

2. Has the statistical analysis been performed appropriately and rigorously?

Reviewer #1: No

Reviewer #2: Yes

Reply: We have included 95% CI with OR in table 4, as suggested by the reviewer.

3. Have the authors made all data underlying the findings in their manuscript fully available?

Reviewer #1: Yes

Reviewer #2: Yes

Reply: Thank you for your comments.

4. Is the manuscript presented in an intelligible fashion and written in standard English?

Reviewer #1: Yes

Reviewer #2: Yes

Reply: Thank you for your comments.

5. Review Comments to the Author

Reviewer #1: Good effort by the investigators. However, there are some major concerns with the methods and analyses that need to be addressed.

1. How were the main outcomes of knowledge and practice adequacy defined? Please provide rationales for choosing the cut-off points for positive attitudes and adequate practice.

Reply: Actually, these are very subjective terms. However, it was considered that if a person has > 70% of the correct knowledge of COVID-19 related questions, then it was considered adequate. In case of attitude, if a person was saying “No” or “not sure” for questions such as “It is my social responsibility to take safety measures in controlling the spread of this infection." It means he/she has a negative attitude, and for four attitude questions, the total score ranged from -4 to +4. The plus scores were taken as positive attitudes, while negative scoring indicated a negative attitude.

Similarly, in the practice section, there were six questions. For practice questions, 2 points were awarded for yes, one for sometimes and zero for no, and the score =>6 scores was taken as adequate and < 6, was taken as inadequate." 

2.What was the response rate? How many medical students were invited to participate vs those who participated in the survey? This has implications on the generalizability of findings to within Pakistan.

Reply: We sent an invitation to different medical universities. A total of 1800 medical students were approached, complete responses were acquired from 1474 medical students with a response rate of 82%. This has been included in the methods. 

3.Please provide 95% CI for the OR in Table 4. As mentioned above, please clarify whether the models have adjusted for any variables.

Reply: In table 4, 95% CI for the OR has been added. The model was not adjusted for any variables.

4. How do the findings about factors associated with COVID-19 related knowledge and practice adequacy in Pakistan compare with literature from elsewhere?

Reply:

5.In the discussion, please add implications of the findings, and what steps are needed to address the gaps identified?

Reply:

Other comments:

Please proofread and correct grammatical errors throughout the paper.

Reply: A professional editing service has done the language editing and a certificate is attached.

Please make the format of the paper consistent (font, color, line spaces) and provide the complete tables.

Reply: The manuscript has been formatted, and tables have been completed as suggested.

Reviewer #2: 

This is an interesting paper and is written at time when it is needed. Furthermore, this manuscript is written on issue which is very relevant in todays world. I have no comment to make hence ask for acceptance.

Reply:

Thank you very much for liking our manuscript.

---

## [Decision Letter · Decision Letter 1]

2 Nov 2020

PONE-D-20-23756R1

Knowledge, attitudes, and practices against the growing threat of COVID-19 among medical students of Pakistan

PLOS ONE

Dear Dr. Noreen,

Thank you for submitting your manuscript to PLOS ONE. After careful consideration, we feel that it has merit but does not fully meet PLOS ONE’s publication criteria as it currently stands. Therefore, we invite you to submit a revised version of the manuscript that addresses the points raised during the review process.

We look forward to receiving your revised manuscript.

Kind regards,

Ramesh Kumar, PhD

Academic Editor

PLOS ONE

Reviewers' comments:

Reviewer's Responses to Questions

**Comments to the Author**

1. If the authors have adequately addressed your comments raised in a previous round of review and you feel that this manuscript is now acceptable for publication, you may indicate that here to bypass the “Comments to the Author” section, enter your conflict of interest statement in the “Confidential to Editor” section, and submit your "Accept" recommendation.

Reviewer #1: All comments have been addressed

Reviewer #2: All comments have been addressed

2. Is the manuscript technically sound, and do the data support the conclusions?

Reviewer #1: Yes

Reviewer #2: (No Response)

3. Has the statistical analysis been performed appropriately and rigorously? 

Reviewer #1: Yes

Reviewer #2: (No Response)

4. Have the authors made all data underlying the findings in their manuscript fully available?

Reviewer #1: Yes

Reviewer #2: (No Response)

5. Is the manuscript presented in an intelligible fashion and written in standard English?

Reviewer #1: Yes

Reviewer #2: (No Response)

6. Review Comments to the Author

Reviewer #1: The responses are satisfactory. However, they need to be reflected in the revised manuscript as well.

Please indicate in your cover letter the page and line number where you made the changes to address the concerns.

Reviewer #2: All comments are addressed by author(s). hence manuscript is in a position to be accepted in PLOS ONE.

7. PLOS authors have the option to publish the peer review history of their article (what does this mean?). If published, this will include your full peer review and any attached files.

Reviewer #1: No

Reviewer #2: No

---

## [Author Response · Author response to Decision Letter 1]

18 Nov 2020

PONE-D-20-23756

Knowledge, Attitudes, and practices against the growing threat of COVID-19 among medical students of Pakistan

PLOS ONE

Dear editor,

Pls find the modified manuscript. We have incorporated all suggestions recommended by the reviewers’. Following is the point-wise reply to the reviewers' comments.

With best regards,

Dr Noreen

Reply to comments

Reply: Rebuttal letter is being uploaded as a separate file.

Reply: The revised manuscript is being uploaded as a separate file as 

Reply: Unmarked manuscript is being uploaded as a separate file.

1.Reply: No changes in financial disclosure 

2. If applicable, we recommend that you deposit your laboratory protocols in protocols.io to enhance the reproducibility of your results. Protocols.io assigns your protocol its own identifier (DOI) so that it can be cited independently in the future

 Reply: Not applicable 

Reviewers' comments:

Reviewer's Responses to Questions

Comments to the Author

1. If the authors have adequately addressed your comments raised in a previous round of review and you feel that this manuscript is now acceptable for publication, you may indicate that here to bypass the “Comments to the Author” section, enter your conflict of interest statement in the “Confidential to Editor” section, and submit your "Accept" recommendation.

Reviewer #1: All comments have been addressed

Reviewer #2: All comments have been addressed

Reply: Thank you for your comments.

2. Is the manuscript technically sound, and do the data support the conclusions?

Reviewer #1: Yes

Reviewer #2: (No Response)

Reply: Thank you for your comments.

3. Has the statistical analysis been performed appropriately and rigorously?

Reviewer #1: Yes

Reviewer #2: (No Response)

Reply: Thank you for your comments.

4. Have the authors made all data underlying the findings in their manuscript fully available?

Reviewer #1: Yes

Reviewer #2: (No Response)

Reply: Thank you for your comments.

5. Is the manuscript presented in an intelligible fashion and written in standard English?

Reviewer #1: Yes

Reviewer #2: (No Response)

Reply: Thank you for your comments.

6. Review Comments to the Author

Reviewer #1: The responses are satisfactory. However, they need to be reflected in the revised manuscript as well.

Please indicate in your cover letter the page and line number where you made the changes to address the concerns.

Reply: Changes made to address the concerns are indicated in cover letter and in which page and line numbers are mentioned. Kindly note that numbering in the document is not continuous because of section breaks. 

Reviewer #2: All comments are addressed by author(s). hence manuscript is in a position to be accepted in PLOS ONE.

Reply: Thank you for your comments.

---

## [Editor Report · Decision Letter 2]

26 Nov 2020

Knowledge, attitudes, and practices against the growing threat of COVID-19 among medical students of Pakistan

PONE-D-20-23756R2

Dear Dr. Noreen,

We’re pleased to inform you that your manuscript has been judged scientifically suitable for publication and will be formally accepted for publication once it meets all outstanding technical requirements.

Kind regards,

Ramesh Kumar, PhD

Academic Editor

PLOS ONE
---

## [Editor Report · Acceptance letter]

4 Dec 2020

PONE-D-20-23756R2 

Knowledge, attitudes, and practices against the growing threat of COVID-19 among medical students of Pakistan 

Dear Dr. Noreen:

I'm pleased to inform you that your manuscript has been deemed suitable for publication in PLOS ONE. Congratulations! Your manuscript is now with our production department. 

Kind regards, 

on behalf of

Dr. Ramesh Kumar 

Academic Editor

PLOS ONE